# Use of a Visual Scoring System to Assess External Udder Conformation and Its Relationship to Colostrum Quality and Lamb Growth Rates

**DOI:** 10.3390/ani13182900

**Published:** 2023-09-13

**Authors:** Haley Richardson, Amin Ahmadzadeh, Denise Konetchy

**Affiliations:** Department Animal, Veterinary, and Food Sciences, University of Idaho, Moscow, ID 83844, USA; rich5704@vandals.uidaho.edu (H.R.); amin@uidaho.edu (A.A.)

**Keywords:** colostrum, ewe, lamb growth, udder conformation

## Abstract

**Simple Summary:**

Because udder shape and milkability are strongly correlated, particular structural traits might be chosen in the future to increase milking effectiveness in sheep. Among udder conformation traits, udder depth and length are most strongly associated with milk output. An efficient selection approach to examine external udder conformation in relation to lamb growth rates and colostrum quality might be produced by a practical method of visual assessments of udder conformation and structure. However, there has not been much research on how the external udder conformation of meat-breed sheep relates to the quality of colostrum and lamb growth rates. The goal of this study was to investigate the relationship among external udder conformation and colostrum quality and lamb growth rates in Suffolk ewes using visual evaluations of the external udder. The study showed that ewes with a ‘normal’ udder conformation had greater mean total protein and Brix values in colostrum compared with ewes with an ‘abnormal’ conformation. Udder conformation based on the parameters of udder floor, udder depth, teat placement, teat lesions, and presence of wool did not appear to have a significant effect on lamb body weight or growth rates. Although udder conformation appears to affect colostrum quality, more research is needed to determine how it affects lamb growth, morbidity, and mortality during the first week of life.

**Abstract:**

In sheep raised for meat production, the relationship between external udder conformation, colostrum quality, and lamb growth rates has not received much attention. We hypothesized that ewes with a more desirable udder conformation at lambing would have greater colostrum quality and greater growth rates in lambs. Fifty Suffolk ewes were used in this study. Within 6–8 h of parturition, colostrum samples from both halves of the udder were collected and visual scoring of the udder was conducted. Colostrum quality was measured for total proteins using both optical and Brix refractometers. On day 2, day 45, and day 60 after parturition, lamb weights were recorded, and udder conformation measurements were repeated. A visual scoring system evaluating udder floor (scale 1–4), udder depth (scale 1–9), teat placement (scale 1–9), teat/mammary lesions (present or absent), and the presence of wool (present or absent) was used to assess the external udder conformation. Normal udder parameters included udder depth scores of 5 or 6; udder floor scores of 1 or 2; teat placement scores of 4, 5, or 6; and the absence of teat/mammary lesions and wool. All ewes not meeting normal parameters were considered to have an abnormal udder. The data were analyzed using the GLM procedure. Mean total colostrum protein was greater (*p* = 0.03) in ewes displaying a ‘normal’ udder conformation compared with those with an ‘abnormal’ conformation (14.82 ± 0.5 and 13.31 ± 0.3 mg/dL, respectively). Mean Brix values were also greater (*p* = 0.03) for ewes with a ‘normal’ udder compared to an abnormal udder confirmation (21.70 ± 0.8 and 19.54 ± 0.5, respectively). On day 2 after parturition, the mean lamb body weight was not different between ewes with ‘normal’ and abnormal udders (5.38 ± 0.26 vs. 5.46 ± 0.15). No differences (*p* > 0.05) in lamb weights were detected between ewes with normal and abnormal udder conformations on day 45 and 60 after parturition. These data provide evidence of greater colostrum total protein values and greater Brix values present in ewes with a ‘normal’ udder conformation. There were no differences in the weights of lambs born to ewes with normal or abnormal udder conformations.

## 1. Introduction

External udder conformation has been widely studied in dairy sheep breeds to determine an effect on milk production and the efficiency of machine milking [1]. Studies have also shown a close relationship between udder conformation and milkability, and that certain conformational traits could be considered for selection to possibly improve milking ability in the future [2]. For example, among udder conformations, udder depth and length were the most correlated traits to milk production, while deep udders and short teats were related to lower somatic cell counts in Spanish Assaf sheep [3]. A study conducted by Zeleke et al. [4] looked at the effect of palpable udder defects on milk yield, somatic cell count (SCC), and milk composition in non-dairy Romney ewes. This study showed, with the exception of the solid non-fat (SNF) composition of milk, no association between the ewe groups and the effects on SCC, milk yield, and milk composition [4].

Interestingly, a study in beef cattle showed that beef calves that had difficulty in suckling because of teat conformation and size showed greater morbidity and mortality [5]. Furthermore, a study in dairy cattle by Ventorp and Michanek [6] showed that calves born to cows or heifers with low-slung udders had difficulty obtaining colostrum in a timely manner by natural suckling, resulting in a negative effect on passive immunity and health in the calf [6].

Different scoring systems have been historically utilized to determine the udder conformation in sheep. These scoring systems classify external conformation traits by using simple numerical scales or physical measurements [1,7]. Measurements have included the udder length, udder width, rear udder depth, udder attachment, cistern depth, teat length, or teat angle.

A study conducted in 2012 was the first that focused on the association between udder conformation and lamb weight in Suffolk ewes by using visual and physical udder measurements [7]. In that study, udder conformation traits of teat placement, teat lesions, and the cross-sectional area of teats were measured two weeks after lambing. It was concluded that teat placement and teat lesions were associated with lamb weight and that the teat conformation in ewes may reduce intramammary infection and increase lamb growth rates [7]. Additional studies conducted by Griffiths et al. [8,9] showed an association between a normal udder conformation and greater mean weaning weights. Furthermore, a normal udder conformation reduced the odds of lamb mortality from birth to weaning [9]. Smith et al. [10] showed an odds ratio for mortality of 1.54 between ewes with unsound udders versus ewes with sound udders. This study identified an odds ratio for mortality caused by starvation of 4.62 between unsound versus sound udders in ewes [10]. Nevertheless, the external udder conformation and its relationship with colostrum quality and growth rates in lambs has not been extensively studied in meat breeds.

High-quality colostrum is essential for the survival of large and small ruminant neonates. Lambs lack the energy reserves necessary for survival and rely on the quantity and protein concentration of colostrum to meet nutritional, metabolic, and immunological needs [11]. According to Nowak and Pascal [11], neonate mortality in sheep production systems is 15 to 25% worldwide, and most preweaning lamb deaths occur within the first week of life. In the study by Griffiths et al. [9], the mortality rate from birth to weaning was 13.8%.

The definition of high-quality colostrum in cows and goats is well documented in the literature, but there is less information available for ewes. One of the most-studied tools for measuring bovine colostrum quality is the Brix refractometer [12]. Many studies have shown the correlation between greater Brix measurements and increased IgG levels [12]. Kessler et al. [12] utilized receiver operating characteristic (ROC) curves to calculate the diagnostic accuracy of the Brix refractometer in the colostrum of cows, does, and ewes. Based on the ROC curve analysis and maximizing sensitivity and specificity, the theoretical optimal cut-off point for Brix in the ewe was 26.5% [12].

A practical method involving visual assessments of the udder floor, udder depth, teat placement, teat lesions, and the presence of wool could result in an effective selection method to evaluate the external udder conformation in relation to lamb growth rates and colostrum quality. The objective of this study was to determine the relationship among external udder conformation, using visual assessments of the external udder, and colostrum quality and lamb growth rates in Suffolk ewes.

## 2. Materials and Methods

### 2.1. Animals

All animals, treatments, and procedures were approved by the University of Idaho Animal Care and Use Committee (IACUC 2022-05). All work was conducted at the University of Idaho Sheep Center. This study used 50 Suffolk ewes between 2 and 5 years of age. The ewes were randomly selected from a group of multiparous ewes by selecting the first 50 ewes to lamb twins. The ewes in the current study were a subgroup of approximately 200 ewes bred to six different Suffolk rams. All ewes in the study were housed outside in a small pasture with continuous access to an adjacent lambing (drop) pen. At night, the ewes were restricted to the drop pen for easier monitoring by the University of Idaho Sheep Center staff, and then again allowed access to the pasture during the day. The close-up (close to lambing) ewes were monitored regularly during the day and every 2 h during the evening and night for lambing. At the sign of labor, trained staff would move the ewe to the drop pen if needed and monitor the ewe during lambing. Once lambing occurred, staff moved the ewe and lambs to a pen (5-foot × 5-foot individual pen) in the adjoining lamb barn. Once in the pen, the lambs’ navels were dipped in an iodine solution, ewe teats stripped to remove any plug, and the lambs were monitored for nursing. The time of birth, sex of lambs, ease of birth, maternal response, and whether any assistance was required were recorded. All close-up ewes were fed a gestation ration with dry matter (DM) adjusted crude protein (CP) of 12.6%, 40% soluble protein, and free access to water. While in the pen, ewes continue to receive the gestation ration, along with grass hay and water. The ewes and lambs were kept in the pen for 72 h before being placed in a mixing pen with 4 to 5 other ewes and lambs and transitioned to a lactation ration. All ewes in the study lambed from 8 March 2022 through 19 March 2022. Study lambs had a creep ration available, starting at 7 days of age from a creep feeder. The creep feeder allowed the lambs to access the ration but not the larger adult ewes. The creep ration consisted of canola meal, rolled corn, rolled barley, dried distillers’ grains, mineral molasses mix, and bypass fat and was offered to the lambs *ad libitum*. External variables such as feed, environment, and management practices were kept consistent for all study animals to limit variability.

### 2.2. Experimental Design and Methodology

A visual assessment was performed on the external characteristics of the udder within 6–8 h of parturition, and on days 2, 45, and 60 after parturition. Most of the lambing occurred between midnight and 04:00 h, and the visual assessment and colostrum collection were done at 07:00 h. Any ewe lambing at other times was evaluated in a timely manner to meet the criteria of the udder evaluation within 6 to 8 h of parturition. All evaluations were conducted by one trained individual. The visual scoring system evaluated the udder floor, udder depth, teat placement, teat/mammary lesions, and presence of wool on the udder. The udder floor (Figure 1) was scored on a 1–4 scale: 1 = defined halving, 2 = too flat, 3 = broken, 4 = asymmetric [2]. Udder depth (Figure 2) was evaluated on a 1–9 scale: 1 = low udder, 9 = shallow udder, 5 = reference point at hock [13]. Teat placement (Figure 3) was scored on a 1–9 scale: 1 = most medial to 9 = most lateral [7,13]. Teat/mammary lesions and the presence of wool were scored as either presence or absence [7]. Each ewe was characterized as having a ‘normal’ or ‘abnormal’ udder based on the following assessments. Normal udder parameters include udder depth scores of 5 or 6; an udder floor score of 1; teat placement scores of 4, 5, and 6; and the absence of teat/mammary lesions, wool, and palpable lesions or masses within the mammary gland. All ewes not meeting any single one of the above normal udder parameters were considered to have abnormal udders. Body condition scores of ewes were recorded when the udder conformation was assessed using the standard 1–5 scale: 1 = emaciated and 5 = extremely fat [14]. Lamb weights were recorded on days 2, 45, and 60 after parturition.

### 2.3. Colostrum and Body Weight

Colostrum samples were collected from both halves of the udder for each ewe within 6–8 h of parturition. Each teat was manually massaged to fill a 15-cc conical tube with colostrum. One drop of colostrum was placed on the optical plate of the optical refractometer (JorVet, Loveland, CO, USA). Total proteins were measured and recorded. Subsequently, the prism was cleaned with distilled water and wiped dry. This procedure was repeated for both halves and the total proteins from each half were averaged to give an overall total protein score for each ewe. A similar procedure was used to analyze each udder half using a Brix refractometer (JorVet, Loveland, CO, USA) to measure Brix values [12]. The Brix values from each half were also averaged to give an overall Brix value for each ewe’s colostrum.

### 2.4. Statistical Analysis

Data on the effects of initial udder conformation on total proteins; Brix values; day 2, 45, and 60 lamb weights; and lamb average body weight daily gain from day 2 to 60 were analyzed using the GLM procedure in SAS, with significance declared at *p* < 0.05 [15]. The model included the fixed effect of udder type (normal vs. abnormal udder conformation) and the random effect of ewe. A repeated-measure generalized linear mixed model [15] was used to determine differences in lamb body weight across time. The model included the fixed effects of udder type, time, and the time-by-udder-type interaction. The initial lamb body weight (day 2) was also included in the analysis as a covariate. Lambs were considered as random effects, and the correlation structure for the repeated measures was ARMA (1, 1).

## 3. Results and Discussion

Based on the initial assessment of the udder conformation, 22% (n = 11) of ewes presented a ‘normal’ conformation and 78% (n = 39) of ewes presented an ‘abnormal’ conformation (Table 1). This initial classification of ewes was used throughout the rest of the experiment to compare mean colostrum total proteins, Brix values, and mean lamb weights between ‘normal’ and ‘abnormal’ udder conformations.

Mean udder floor scores differed (*p* < 0.01) between the two udder types and were 1 ± 0.4 and 2.5 ± 0.2 for ewes with normal and abnormal udders, respectively (Table 1). Furthermore, the udder depth scores were greater (*p* < 0.05) for ewes with abnormal udders than those with normal udders (6.4 ± 0.2 vs. 5.6 ± 0.3). Mean teat placement scores tended (*p* = 0.08) to be lower for ewes with normal udders compared with ewes with abnormal udders. Overall, 33.3% of ewes with an abnormal udder had a teat placement score > 6 and no ewes had teat placement scores < 4 (Table 1).

The mean BCS of ewes averaged 2.61 and was not different (*p* = 0.80) between the two udder types (Table 2). Mean total colostrum protein was greater (*p* = 0.02) in ewes with a normal udder conformation (14.82 ± 0.58 mg/dL) compared with ewes having an abnormal conformation (13.31 ± 0.33 mg/dL) (Figure 4). Brix values were also greater (*p* = 0.03) in ewes presenting a normal conformation (21.70 ± 0.88) compared to ewes presenting an abnormal conformation (19.54 ± 0.52) (Table 2).

The results demonstrate that a normal udder conformation may be associated with greater total protein amounts in the colostrum. Brix values have been highly correlated to IgG in ewe colostrum, with both frozen and thawed samples [12,16]. The Brix values in the current study were lower than the theoretical optimal cut-off point determined by Kessler et al. [12] of 26.5% but within the range for does at 21%. Santiago et al. [17] found a significant drop in ewe colostrum proteins between 6 and 12 h post-lambing, most likely due to colostrum consumption by the lambs. Therefore, our criteria of evaluation 6 to 8 h post-lambing and after lambs were confirmed to have nursed may have contributed to the lower Brix values. The colostrum and milk composition differ among different sheep breeds [17]. Others have observed a wide range of IgG concentrations in sheep (6.2 to 65.4 mg/mL), with greater colostral IgG concentrations in meat-type compared with milk-type breeds [18]. Nevertheless, data are lacking on IgG concentrations in meat breeds such as the Suffolk breed.

The greater total protein and Brix values of the colostrum exhibited in the normal udder conformation may be a result of better intramammary colostrogenesis. It can be speculated that a normal udder conformation may have advantages in terms of the size and structure of secretory lobules or alveoli, allowing for the greater production of colostrum components including proteins, resulting in better-quality colostrum. Barbagianni et al. [19] found that the size of gland cisterns correlated with milk production in dairy sheep, as ewes exhibiting larger gland cisterns resulted in greater milk yields. However, colostrum quality was not reported in the study, yet the larger gland cisterns may potentially contribute to greater colostrum production. Another study by Caja et al. [20] observed that the size of the gland cistern varied depending on the breed and milking ability, with dairy breeds exhibiting larger gland cisterns compared with non-dairy breeds. However, the gland cistern size would most likely affect the amount of colostrum and milk that is able to be produced rather than the quality of colostrum.

Abecia et al. [21] found a relationship between lamb sex and IgG concentrations. Ewes with a ram lamb in the litter had greater IgG levels in their colostrum compared with ewes with no rams in the litter [19]. In the current study, the sexes of the lambs between both groups were as follows. In the normal udder ewe group, 72% of ewes had a twin that was a ram. None of the ewes in this group had twin ram lambs and the remaining (28%) had twin ewe lambs. In the abnormal udder group, 14% of the ewes had twin ram lambs, 55% had a combination of ewe and ram lambs, and 31% had twin ewe lambs. Although the percentage of ram lambs was numerically greater in normal udder ewes than abnormal (72 vs. 69%), because of the limited number of observations, it was difficult to establish a relationship between lamb sex and the total colostrum protein in the current study.

The importance of high-quality colostrum should not be overlooked, with most preweaning lamb deaths occurring within the first week of life and a worldwide neonate mortality rate in sheep production systems of 15–25% [9]. In the current study, we saw a 3% birth-to-weaning mortality rate in lambs (all lambs died from birth to 72 h and were from ewes in the abnormal udder group). This differs from the study of Griffiths et al. [9], where they saw 13.8% mortality in lambs from birth to weaning. The observed differences may be related to the greater number of lambs in their study [9] compared with the current study, as well as management. The study of Griffiths et al. [9] managed lambing in a pasture situation, whereas the current study managed the ewes in a more intense lambing management situation, providing more intense observations of ewes and lambs along with feeding and colostrum management. Nevertheless, the quality of colostrum should be emphasized along with the quantity of colostrum in sheep production systems to ensure the survival and vitality of lambs. This study suggests that by evaluating the external udder conformation in ewes, we can infer the relative quality of colostrum being produced to better ensure the survival of the lambs. Practical management practices can potentially be implemented based on the evaluation of the external udder conformation. For example, excess colostrum could be collected from ewes with better udder conformations to be frozen for later use or fed to lambs with a dam exhibiting a poor udder conformation. This could be an important management strategy to identify ewes producing high-quality colostrum by evaluating the udder conformation.

There was no effect of udder conformation or udder-conformation-by-day on the body weight of the lamb. However, as expected, there was an effect of day on lamb body weight (*p* < 0.01). There was no difference (*p* > 0.3) in mean lamb weights at day 2 (2.44 ± 0.12 kg vs. 2.48 ± 0.07 kg), day 45 (17.28 ± 0.91 kg vs. 16.02 ± 0.60), and day 60 (22.5 ± 1.31 kg vs. 21.1 ± 0.68 kg) postpartum between the normal and abnormal udder conformations (Table 3). Furthermore, the lamb average daily gain from day 2 to day 60 showed no difference (*p* = 0.33) between the normal (0.75 ± 0.04 kg) conformation and abnormal (0.70 ± 0.02 kg) conformation (Table 3). These results demonstrate no differences in lamb average daily gain between ewes with normal and abnormal udder conformations. These findings are not consistent with a previous study [7], which demonstrated that the offspring of ewes with more desirable teat placement and fewer teat lesions on week 2 after parturition had greater rates of lamb growth. Another study by Griffiths et al. [8] demonstrated a 2.1 kg lower mean weaning weight in lambs from ewes with abnormal udder scores compared with ewes with normal udder scores. The difference in outcomes may be reflected by the different study methods; Griffiths et al.’s [8] study enrolled 1570 lambs, with a mean weaning age of 84.4 days and no access to creep feed [8], while the current study enrolled 100 lambs, with a mean weaning age of 60 days with access to creep feed *ad libitum*.

Huntley et al. [7] reported that an excellent teat conformation, including teat placement scores of 5 and a greater cross-sectional area of the teats, reduced intramammary infection. This may be because of the greater volume of residual milk resulting from a larger teat cistern or because the teats have a more secure teat sphincter, reducing bacterial entry [7]. When considering the teat placement, a placement score of 5 suggests that this score is the optimal position to allow lambs to effectively suckle, based on an earlier study [7]. As there was no difference in lamb growth rates between the normal and abnormal conformation groups, we can deduce that the external udder traits of udder floor, udder depth, and presence of wool may have no significant effect on lamb growth rates.

To our knowledge, the information on the relationship between palpable udder defects and colostrum quality is lacking. In the current study, only two ewes exhibited palpable udder defects and were considered ewes with an abnormal udder. Whether palpable udder defects alone are associated with colostrum quality could not be determined from the current study. Zeleke et al. [4] only investigated the effect of palpable udder defects on milk yield, somatic cell count (SCC), and milk composition in non-dairy Romney ewes and found no association between palpable udder defects and SCC, milk yield, and milk composition [4]. Others [9,10] found that lambs born to ewes with unsound udders are at an increased risk of neonatal mortality. Nevertheless, none of these studies [4,9,10] evaluated the relationship between udder soundness and colostrum quality.

As previously noted, the current study lambs had a creep ration available, starting at 7 days of age. The creep ration consisted of canola meal, rolled corn, rolled barley, dried distillers’ grains, mineral molasses mix, and bypass fat and was offered to the lambs’ *ad libitum*. Providing a creep ration may have had an impact on the ADG between the two udder conformation groups. If lambs were not receiving enough nutrients from the dam, there was another source available to account for this deficiency. All lambs had continuous access to the creep ration, potentially resulting in no difference in ADG between the conformation groups. This study found no differences in lamb growth rates between udder conformations; although previous studies found a correlation between lamb growth and teat placement in Suffolk sheep [7], a creep feed was not available to the lambs in this study. Again, the creep-feeding management of the lambs in this study may have factored into the lamb growth rates by providing an additional food source for the lambs other than lactation.

Additional research may be needed to further evaluate the relationship between lamb growth and udder conformation, specifically in meat breeds. Research on lamb carcass values in the context of udder conformation may reveal an influence of high-quality colostrum and udder conformations on carcass traits. The genetic influence of the udder conformation on colostrum values and growth rates has also yet to be considered.

## 4. Conclusions

By identifying ewes that produce higher-quality colostrum on the basis of the udder conformation, producers could better support lamb survival and potentially reduce lamb mortality even prior to lambing. Additionally, production management practices such as collecting and freezing colostrum from ewes with a normal udder conformation may increase the overall quality of stored colostrum in a production system. Although udder conformation appears to influence colostrum quality, further research evaluating the relationship between udder conformation and lamb mortality and morbidity in the first week could have implications for both lamb and ewe management.

## Figures and Tables

**Figure 1 animals-13-02900-f001:**
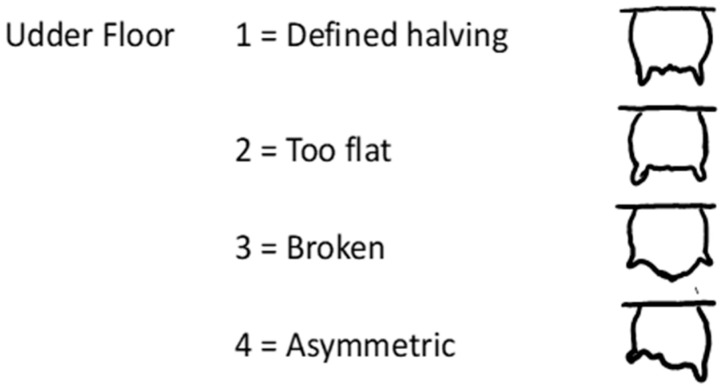
Scoring system used to assess udder floor consisting of defined halving (1), too flat (2), broken (3), and asymmetric (4) [2].

**Figure 2 animals-13-02900-f002:**
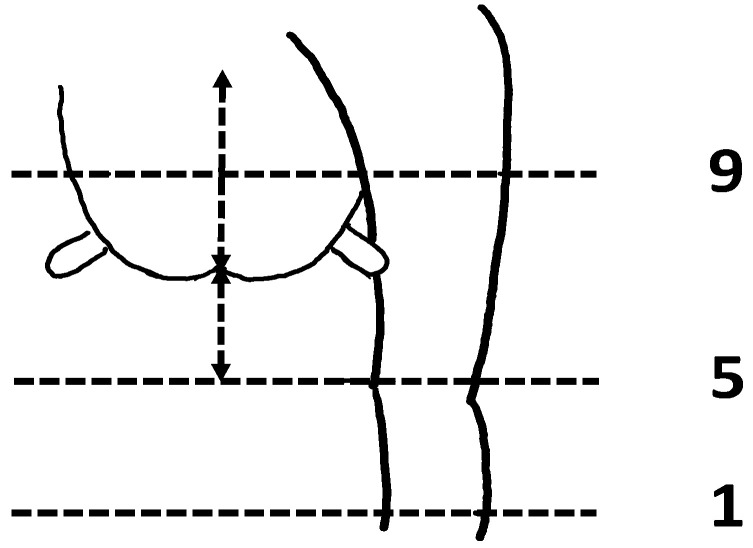
Nine-point scoring system used to evaluate udder depth from shallow (9) to low (1), with the udder floor at hock (score 5) as a reference point [13].

**Figure 3 animals-13-02900-f003:**
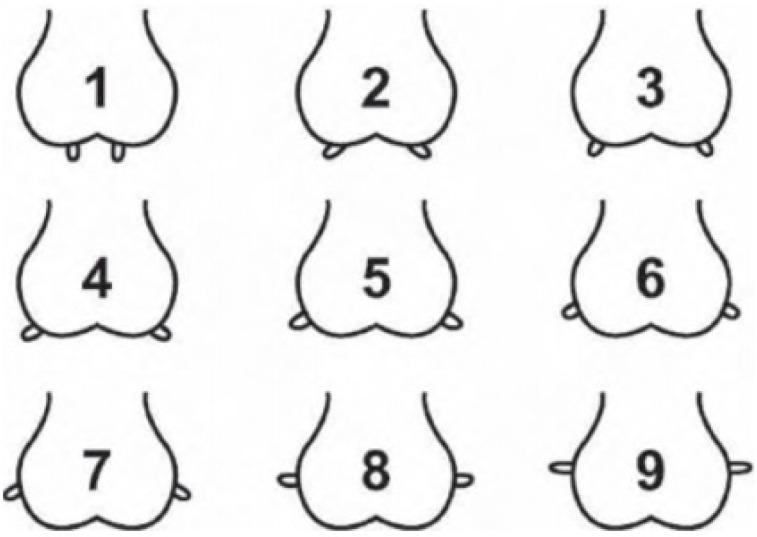
Nine-point scoring system used to evaluate teat placement from most medial (score 1) to most lateral (score 9) [6,7].

**Figure 4 animals-13-02900-f004:**
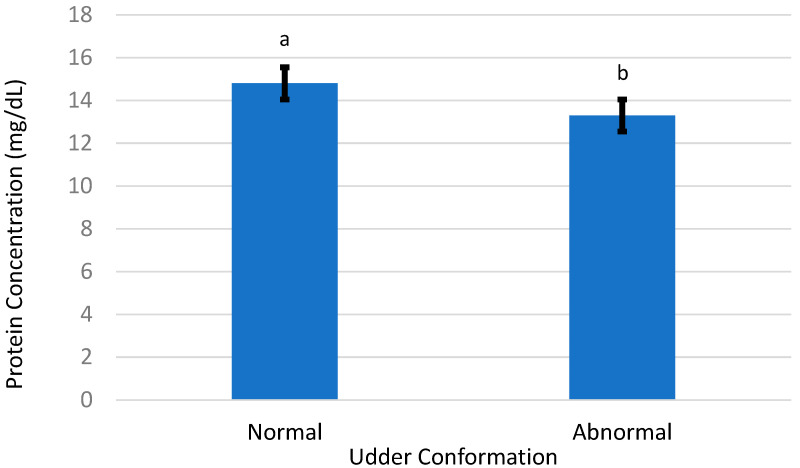
Mean colostrum total protein from Suffolk ewes with ‘normal’ and ‘abnormal’ udder conformations; a,b, bars with different letters differ (*p* = 0.03).

**Table 1 animals-13-02900-t001:** Mean (±SEM) udder floor, udder depth, and teat placement scores of ewes with normal and abnormal udder conformation.

Items	Udder Conformation Type		*p* Value
	Normal (n = 11)	Abnormal (n = 39)	
Udder floor	1.0 ± 0.3 ^a^	2.6 ± 0.2 ^b^	0.01
Udder depth	5.5 ± 0.3 ^a^	6.4 ± 0.2 ^b^	0.04
Teat placement	5.3 ± 0.3 ^c^	6.0 ± 0.2 ^d^	0.08

^a,b^ Means with different superscript within the row differ (*p* < 0.05). ^c,d^ Means with different superscript within the row tended to differ (*p* < 0.1).

**Table 2 animals-13-02900-t002:** Least square means (±SEM) of body condition scores (BCS) ^1^ and Brix values in two udder conformation types in lactating ewes.

Variable	Udder Conformation Type		*p* Value
	Normal (n = 11)	Abnormal (n = 39)	
BCS	2.60 ± 0.14	2.63 ± 0.08	0.80
Brix%	21.70 ± 0.88 ^a^	19.54 ± 0.52 ^b^	0.03

^1^ Body condition scores of ewes were recorded using the standard 1–5 scale: 1 = emaciated and 5 = extremely fat [9]. ^a,b^ Means with different superscript within the row differ (*p* < 0.05).

**Table 3 animals-13-02900-t003:** Mean (±SEM) lamb body weight on days 2, 45, and 60 postpartum and average daily gain (ADG) of lambs from ewes with normal and abnormal udder conformation.

Age of Lamb	Udder Conformation Type		*p* Value
	Normal (n = 11)	Abnormal (n = 39)	
Day 2	2.44 ± 0.85	2.48 ± 0.48	0.77
Day 45	16.65 ± 0.98	16.01 ± 0.51	0.43
Day 60	22.44 ± 0.98	21.10 ± 0.51	0.43
ADG	0.75 ± 0.04	0.70 ± 0.02	0.33

## Data Availability

Not applicable.

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
