# Peer review of "Use of a Visual Scoring System to Assess External Udder Conformation and Its Relationship to Colostrum Quality and Lamb Growth Rates"

_animals, 2023, doi:10.3390/ani13182900_

Round 1

Reviewer 1 Report

This is a well performed, well written study, and is useful to breeders and investigators, even if the study was largely negative w.r.t the aims.

Minor comments and suggested edits follow.

L38        Insert ‘respectively’

L60        Comma after suckling

L89        Were there any difference in udder scores that could be related to sire?

L94        Comma after labor

L96        What is a jug?

L97        What happened to the stripped material? Is this colostrum?  Was any mammary content stripped as well?

L123      Extra full stop

L139      By 6 hours, would the lamb have already consumed the best of the colostrum?

L148      What is the editorial policy on the use of abbreviations?

L161      Should other be udder and if not, and what particular types?

L174      Figure 4, P values are often included on the plot, either as values or as asterisks

L179      0.2 b (extra space)

L187-202            Agreed, also an indication of low disease

L207      and after lambs

L218      As there were no lamb losses, I’m not sure that the next sentence is justified, suggest replacing Therefore with Nevertheless

L239-241            I don’t see how extra measures per se would account for differences in lamb growth rates.

L258      Providing the creep ration may have undermined the aim of the study, but good that this was mentioned and discussed.

L325      Larger font used.

The use of English is very good. Few changes suggested.

Reviewer 2 Report

Use of a visual scoring system to assess external udder conformation and its relationship to colostrum quality and lamb growth rates.

General

An interesting study on a relevant topic. Well done on investigating this and adding to the information available. However, missed opportunities to better describe and investigate the data collected (see below for further detail). Too much speculation in the discussion about things outside of this study. Some recent literature in the field missed.

Introduction

Lines 59-60 “cannot be expected to obtain colostrum soon enough” is poorly worded – suggest re-phrasing.

Lines 67-74 – suggest reordering this paragraph to make it clear to the reader at the outset that there is limited literature available for meat breed ewes with regards to udder conformation. Although, note there has been some recent (2019 – 2023) work done on udder health (including conformation) in meat breed sheep in New Zealand & Australia that will be useful for this manuscript and should ideally be included.

Methods

Line 86 – how were some of the multiparous ewes 1? Even if they were bred as ewe lambs, they still would have been 1-year-old at time of their first lamb(s) being born, which would make them ~2 when their second lamb(s) are born?

Line 89 – which breed of ram(s)?

Line 90 – space before brackets. Appreciate the clarification as to a ‘drop’ pen. Suggest providing clarifications for terminology throughout as differences between countries e.g., jug, mixing pen, close up.

Line 122 – extra "."

Line 122 – 123 – you mention palpable masses here but no description of how you assessed these and classified them (and how this compares to other literature that looked at udder palpations). This should be described please.  

Since you had milk samples, did you consider assessing for clinical or subclinical mastitis – this kind of information is lacking in meat breed sheep and would be really interesting. it would also add additional value to your work. 

Collapsing into normal vs abnormal seems like it will miss differences or potential for some characteristics to be more or less important than others? I think a reader will want to see the number of ewes in each category and why they were considered abnormal. The ewes were assessed 4 times (lambing, day 2, 45 & 60) – but it’s unclear which score(s) were used in the analyses – presumably not all ewes were the same score in all categories at all 4 time points? I think this could be explored further since you have the data too –a missed opportunity? Obvs not sensible for colostrum measures (chronological order) but surely for lamb parameters? Presenting this data would be very useful given the lack of data available in this field. Edit - I see when I got to the results you explain this a little more, but the points around additional detail and value remain. Important to see number of ewes with each score. 

Line 137 onwards - What happened if you had ewes with large variation between halves e.g., one side mastitis or one side agalactia? Was this able to be accounted for in your methods?

Did all 100 lambs survive? It seems unlikely that there was not a single lamb mortality in the study given typical mortality rates in sheep flocks globally. It would be good to know the number of ewe-lamb pairs used in the final analyses as this could have been a confounder for your results?

Pre-lambing vaccinations given? If so, what & when? (potential for impact on colostrum)

Line 148 – please describe the statistical methods more clearly. How was model fit assessed? Data processing/assessment before modelling? What are the outputs of your stats?

Results & discussion

Line 156 – if all based on the initial assessment then why did you bother with assessments at the other time points?

Line 166 – the actual number of ewes in each score aren’t reported anywhere – suggest they should be rather than just comparing means. Adds a lot more value and robustness. 

Line 173 – formatting, no spaces in results.

Table 1 – doesn’t make sense. Why do you need superscripts when there are only 2 categories ‘normal’ and ‘abnormal’ and a simple p-value reported?

Table 2 – is this a complete table? The descriptions don’t match? Why now talking about body weight? (lines 184 – 185)

Figure 4 and Table 2 have the same information – don’t need both

Line 188 – 189 – be useful for reader to have an idea of what is considered ‘optimal’

Lines 214 – 215 – what is the recommendations – more useful for reader to have the actual numbers than vague terms

Line 217 – if you had no lamb deaths, why now discussing lamb mortality with relation to the udder conformation – you don’t have data to support this from your study? I think the NZ work looked at lamb survival too - might be useful here for this argument. 

Paragraph 228 – I think the NZ work looked at lamb growth rates too – please check.  

Lines 255 – 257 – could this be due to simplifying the scores?

Conclusion

Lines 272 – 278 seems too speculative to have as a conclusion when not directly based on this study. 

Lines 284 – 288 – this is the first this has been mentioned and is not relevant to the paper – please remove – or if you wish to discuss these, they should be in the discussion, not the conclusion.

N/A 

Reviewer 3 Report

see comments in the attached 

Round 2

Reviewer 2 Report

Line 73 & 83 - you've spelt the author's name wrong (Griffiths not Griffith) 

There are references that should ideally be included that aren't as both are very relevant to this manuscript (one of the other reviewers also pointed this missing literature out in the 1st round of reviews) - https://doi.org/10.3390/ani11102831 & https://doi.org/10.1016/j.smallrumres.2023.107019

Line 204 - there is no 'd' in the Table, what is this in reference to? 

"Given this is a “Brief Report” manuscript, and the fact the udders were categorized into two types, we do not think this information adds to this “Brief Report”." - I still feel it would be useful to include this, even as an appendix but will not force this if the authors and editor are Ok to leave out. 
